# Low Motor Dexterity and Significant Behaviors Following Hospitalized Isolation in Children

**DOI:** 10.3390/children10081287

**Published:** 2023-07-26

**Authors:** Kaitlin Fraser, Miriam Kuhn, Rebecca Swanson, Don W. Coulter, Christopher Copeland, Jorge M. Zuniga

**Affiliations:** 1Department of Biomechanics, University of Nebraska Omaha, Omaha, NE 68182, USAccopeland@unomaha.edu (C.C.); 2Department of Special Education and Communication Disorders, University of Nebraska Omaha, Omaha, NE 68182, USA; 3Pediatric Hematology and Oncology Clinic, University of Nebraska Medical Center, Omaha, NE 68198, USA; 4Division of Pediatric Hematology and Oncology, University of Nebraska Medical Center, Omaha, NE 68198, USA; 5Children’s Hospital & Medical Center Specialty Pediatric Center, Omaha, NE 68114, USA

**Keywords:** patient isolation, motor skills, brain imaging, behavioral rating scale, adolescent health, fNIRS, motor cortex

## Abstract

The main objective of this study was to describe the cortical patterns of brain activity during a gross dexterity task and develop a behavioral profile of children experiencing isolation. A cross-sectional assessment was conducted during one visit. Sample: Four pediatric patients who had undergone isolation within a hospital comprised the full data collection. During the collection, participants completed the Box and Blocks Test of gross manual dexterity while undergoing imaging of the motor cortex using functional near-infrared spectroscopy. Participants also completed a Behavioral Assessment System for Children, Third Edition (BASC-3) self-report, which was analyzed along with a parent report to quantify their emotional and social behaviors. All participants displayed lower gross dexterity levels than normative data. Furthermore, three out of the four participants displayed ipsilateral dominance of the motor cortex during the dexterity task. Three of the participants displayed behavioral measures reported within clinically significant or at-risk scores. Clinically significant behavioral scores coupled with lower than expected manual dexterity values and ipsilateral hemispheric dominance indicate that neuroplastic changes can occur in populations undergoing hospitalized isolation. While the impacts of the treatments and isolation in this case cannot be separated, further studies should be conducted to understand these impacts of isolation.

## 1. Introduction

In the United States, children are often placed in protective hospitalized isolation periods to protect their health and allow for the completion of their therapeutic regime. To treat cancer, children are often placed on a therapeutic regimen involving pharmaceuticals to suppress their immune system [1]. Children undergoing whole organ transplants will also undergo immunosuppressive therapy to diminish the risk of new tissue rejection [2]. During the time that these children are immunosuppressed, they commonly avoid public areas where they might be exposed to pathogens. They may also be confined within a hospital for portions of their immunosuppression treatment [3]. Children with chronic illnesses who are at an increased risk for acquiring multidrug-resistant organisms are also likely to be placed in protective hospital isolation [4]. Hospitalized isolation varies from social isolation due to strict guidelines regarding the number of visitors which are enforced by hospital guidelines. These hospital guidelines vary, but during the COVID-19 pandemic, pediatric patients were frequently limited to visitation by parents alone [5]. However, the arising effects due to hospitalized isolation remain unclear.

While hospitalized isolation can be used to protect children from infections, it can also prevent the experiences gained through socialization, which are important for development. Children with chronic illnesses are less likely to be physically active and to be able to experience a wide variety of social interactions than their healthy peers, which can be amplified while under hospitalized isolation [6]. Separation due to the risk of infection could lead to longer-term deficits in attachment with others [5,7]. Due to these factors, they frequently miss out on the beneficial aspects of physical activity and social interactions with their peers gained through socialization, which frequently occurs through play behaviors [8,9,10]. This paucity of these play behaviors may lead to deleterious long-term consequences [11]. Piaget’s theory of development and the Neuronal Group Selection theory both predict that development builds on previous stages that are influenced by both genetics and experiences [10,12]. Although the beginning stages of development are more driven by genetic factors, the experiential influence increases through the child’s development. According to these theories, children in isolation would be expected to display a differential development of their movement patterns and neural pathways due to their relative lack of varied experiences [10,12]. Based on both Piaget’s theory and the theory of neuronal group selection, the lower practice levels of gross dexterity tasks would increase the complexity of the task, which could additionally lead to a differential activation of the motor cortex. In humans, motor cortices have displayed an increased ipsilateral dominance when performing more complex tasks, such that the left side of the motor cortex is more active than the right side when controlling the left hand and vice versa [13]. This cortical difference due to isolation has been supported in previous studies that have been conducted using animal models, which have identified the presence of neurological differences within the brain after isolation [14,15,16,17,18,19]. When better understood, these effects, which can be quantified through non-invasive functional brain imaging, could be utilized to quantify and modify the interventions addressing the impacts of isolation as the patients actively experience it. However, the specific neurological impacts of isolation in children have not been well researched.

Studies dating back over fifty years have examined the impacts of isolation on children’s behavior and cognitive development, but there remains a lack of quantitative research into these impacts. Previous studies have focused on qualitatively assessing differential development through descriptions of how children complete activities or through structured interviews [20,21,22], but this limits the opportunities for the replication of the results and comparisons between different research groups. This is particularly important for informing specific hospital policies surrounding the isolation of pediatric patients. In these qualitative studies, children frequently reported depressive thoughts and anxiety following isolation, but the impacts and reversal of their isolation were variable [20,22,23,24,25,26,27]. While short-term reversal could be observed, future traumas are more likely to impact pediatric patients, leading to longer-term impacts [28]. To aid in the development and implementation of the policies and interventions to limit the short- and long-term negative impacts of isolation, more information is needed to understand what specific developmental behavioral and cognitive areas are impacted by isolation.

The current study was designed to quantify the impacts of isolation in chronically ill children on their motor skills, behavior, and neural activity. Due to an anticipated lack of motor experience and associated neural changes predicted by the theory of neuronal group selection and Piaget’s theory of development, the patients in this study were expected to display lower levels of motor dexterity compared to normative values that were established in typically developing children. They were also expected to exhibit an ipsilateral-dominated activation of the motor cortex during motor tasks. Additionally, in agreement with previous qualitative studies, participants were expected to report levels of behaviors indicating emotional and social development which would be clinically relevant compared to the frequency with which they are reported across age- and sex-matched cohorts.

## 2. Materials and Methods

### 2.1. Design

The current experiment is a retrospective observational study into the impacts of hospitalized isolation. Data are presented for each participant individually due to the many factors which may contribute to these findings. 

### 2.2. Participants

Six participants were recruited for the study and were granted informed consent as approved by the University of Nebraska Medical Center Institutional Review Board. All experiments were conducted in compliance with the guidelines and regulations outlined by the relevant institutional review board. All participants had undergone hospitalized isolation and were either in continued hospitalized isolation or social isolation at the time of the data collection. All of the participants’ parents granted informed consent and all participants provided their informed assent. The parents were sent a copy of the informed consent form and information sheet during the scheduling of the data collection to allow as much time as possible to review the experiment and ask questions, in large part, because of the highly anticipated levels of anxiety surrounding the medical procedures to be performed by their children. Parents had the opportunity to email or call with questions throughout the process and were present in the room throughout the process of data collection. Two of the participants (S1 and S2) were excluded from the data analysis due to incomplete datasets of functional brain imaging data and gross motor dexterity data.

Data for each participant were recorded during a single visit to the hospital. To avoid an unnecessary risk of exposure to pathogens, these visits were scheduled to coincide with a pre-scheduled routine visit to a clinician within the University of Nebraska Medical Center. At the time of data collection, precautions at the hospital against COVID-19 included a temperature screening process prior to entry and the restriction of only one guest per patient. Accordingly, each participant was accompanied by only one parent, and the number of members of the research team was limited as much as possible for each collection. Parents also answered health screening questions approved by the ethics board prior to commencing the collections over the phone. Throughout the data collection, researchers wore masks, eye protection, and disinfected surfaces and their hands in compliance with the hospital and institutional review board policies. Additionally, steps were taken throughout the data collection to guarantee the comfort and protection, both physically and emotionally, of the participants. These steps will be detailed throughout this section.

### 2.3. Instruments and Variables

In brief, the data collection was designed to assess motor control, their associated neuroplastic adaptations, and behavioral indicators of the emotional categories. For each participant, a parent completed a questionnaire regarding the cumulative number and length of stays in isolation throughout the participants’ lives. Personal information, such as age, sex, and dominant hand, were also recorded. The participants completed three trials of the Box and Blocks Test of gross manual dexterity on each hand while their brain activation patterns were being recorded [29,30]. The participants and their parents then completed the BASC-3 behavioral assessment as appropriate based on the child’s age.

### 2.4. Procedure

The Box and Blocks task has commonly been used across rehabilitation settings to quantify gross motor dexterity [29]. It is also an easily portable test which can be transported to patient rooms with little effort to monitor patients under hospitalized isolation. The participants were given one minute for each trial to move as many blocks as possible from one side of the divided box to the other. For this study, the equipment was 3D printed on an antimicrobial PLA filament embedded with copper particles to increase the safety of the participants. To increase the level of comfort with the researchers and the experimental setup the participants were told that the blocks were 3D printed on this filament and were given the opportunity to ask questions or play with the blocks while the brain imaging equipment was being sized to the participant. Prior to the commencing the task, participants were asked to practice the task to demonstrate an understanding of the instructions. If the participants move more than one block at a time, only one block was counted. Throughout the test, the participants were seated in a stationary chair at a height comfortable for them to only move their arms and hands to complete the task [29]. If necessary, cushions were added to the seat and under the participants’ feet to facilitate this position. Between each trial, the participants were asked to remain motionless for one minute to allow their hemodynamic activity to return to baseline in accordance with previous studies [31]. Participants were informed of their scores after each trial to maintain motivation throughout the task by attempting to beat their own previous scores. Three trials were completed by each participant on each hand consecutively, with the beginning hand randomized for each data collection.

Through the one-minute dexterity task, the participants’ cortical hemodynamic activity was assessed with measured with a continuous wave fNIRS system (NIRSport 2, NIRx Medical Technologies, LLC, Berlin, Germany). This system was chosen due to the portability of the fNIRS system and its relative tolerance for noise due to movements. The NIRSport 2 was placed on the back of the participants’ chair with a harness to allow a more comfortable seated position for the participant. The probes consisted of 8 sources which were monitored by 8 detectors that were spaced 3 cm apart. The spaces between each source and the surrounding detectors were referred to as the channels. The probes with channels above the C3 and C4 motor cortex were placed following the montage designed by NIRx to allow for the monitoring of the hand motor regions and for the subsequent acquisition with the Aurora fNIRS system (NIRx Medical Technologies) [31]. The head circumference of the participants was obtained after informed consent was gathered. To ensure good contact between the probe and scalp, the stretchable cap was chosen to be the closest size below the circumference measurement. This is in compliance with the recommendations outlined by NIRx Medical Technologies to ensure high-quality data collection. In the case of participants who did not have hair, they were given the option to size up the cap to allow for a slightly looser fit. This choice was offered in part due to a high sensitivity to discomfort from the probes and to a lack of hair which would disrupt the path of light. In all cases, the fNIRS cap and probes were covered with a sanitized shower cap. To further control for the impact of lighting, all data collection methods were completed in the same conference room with the fNIRS system commencing in a corner far from the skylight present in the room. The overhead lights were adjusted when necessary to ensure that the levels of light remained consistent between participants. 

Previous studies with pediatric patients who have undergone hospitalized isolation have reported a high level of anxiety with healthcare equipment and a sense of punishment associated with their experience [25]. However, the play-based introduction of new procedures, such as giving pretend shots to a teddy bear and having the child “administer” the treatment, has been proven to be effective at reducing anxiety [32]. Accordingly, participants were encouraged to help place fNIRS caps on a researcher prior to their own cap placement, with the researcher also wearing the cap throughout the time that the child wore their instrumented cap to increase the child’s enjoyment of the process. Additionally, participants were shown how to place a probe in the cap and were permitted to touch the non-light-emitting portion of the probe to grow more familiar with the apparatus prior to placement on their scalp. The cap was not placed on the participants’ heads until they expressed their comfort with the setup. These familiarization procedures were repeated when necessary prior to the placement of the cap. To maintain motivation during the process of data collection, participants were told that they would be given the opportunity to view a live depiction of their hemodynamic activity following the completion of the acquisition within the Aurora software. Care was taken to explain all procedures at an age-appropriate level. 

The BASC-3 assessment was designed and validated to assess self and parent reports of behavior which can indicate emotional and social development [33]. It is also a tool that is widely used by school counselors to monitor the behavioral indicators of children. It includes questions that were designed to evaluate the responses to multiple questions to gain insights into specific types of behaviors that can indicate emotional and social development [33]. These forms are specific to different age ranges. The parent forms are validated for children aged 2–5 years, 6–11 years, and 12–21 years, respectively. The self-report forms are validated for children aged 8–11 years and 12–21 years, respectively. As appropriate based on the age parameters of the forms, each participant was asked to complete the self-report form, while their parent was asked to complete the parent report form [33]. The participants were asked to complete the questionnaires as honestly and completely as possible. Trained personnel were present in the room to answer questions from the participants as they completed the questionnaire in compliance with proctoring guidelines.

### 2.5. Data Analysis

Changes in oxygenated hemoglobin (HbO) were calculated through changes in the optical density that were obtained using the modified Beer–Lambert Law [34]. Data were analyzed in the AnalyzIR Toolbox [35] in MatLAB (The MathWorks, Inc., Natick, MA, USA). This package allows for the analysis of time series fNIRS data through general linear models [35]. Differences from the predicted data were used to calculate t statistics for each channel. The alpha level was set at 0.05 following a Benjamini–Hochberg correction. The laterality index (*LI*) was calculated to assess hemispheric dominance using Equation (1) [31,35]:(1)$$LI=\frac{Oxy_{l}-Oxy_{r}}{Oxy_{l}+Oxy_{r}},$$
where *l* represents the channels from the left hemisphere and *r* indicates the channels from the right hemisphere, respectively. Using this equation, left hemisphere dominance would include the values from 0.2–1, with the right hemisphere including their corresponding negative values. Values from −0.2 to 0.2, respectively, were considered as indicative of bilateral dominance [36]. As this equation is a ratio and uses averaged HbO values from the channels in the regions of interest, the values are unitless and do not incorporate standard deviations in the results presentation.

After coding the BASC-3 results, T scores were assigned to each behavioral outcome based on the amount of age- and sex-matched peers who have indicated similar behavioral outcomes nationally. The T scores were then used to assess the scales falling within the at-risk and clinically relevant ranges, as outlined by the designers of the assessment [33].

## 3. Results

All participants included in the data analysis were right-handed, male, 9–16 years old, and had undergone hospitalized isolation. The length of stay varied from 2 days to 27 days, respectively, and was due to different reasons for each participant (as detailed in Table 1). Due to the heterogeneity of the group and the small sample size, these results are presented individually by participant to allow for a better understanding of the unique characteristics of these participants.

Hemodynamic activity was monitored during the completion of the gross motor dexterity task to ensure that the results can be reported together to allow for a comparison to be made between the hemodynamic activity of a participant with their corresponding functional outcomes. The results of the gross motor dexterity task were reported based on the averaged number of blocks moved unimanually across the three trials performed during the one-minute task. The participants in the current study moved averages of blocks per minute ranging from 43.5–74.7% compared to the normative values of blocks established by Mathiowetz et al. during their trials (Figure 1) [29]. The average difference between the expected and obtained results across the participants and their hands were 32.39 fewer blocks obtained. The participants individually moved anywhere from 19 blocks to 43 blocks fewer than expected, respectively, compared to previously published normative figures for their age and sex (Figure 1). The performance between hands varied, with subject 3 moving fewer blocks with his right hand than his left, but the remaining participants each moved more with their dominant right hands. Figure 2 displays the heterogeneous distribution of dominant hemispheres, which differs from the previously published results involving typically developing pediatric populations. The laterality values were not found to be statistically significant but may present crucial clinical implications. Of the four participants, two displayed ipsilateral dominance during movements with one hand (S5 with their left hand, and S6 with their right hand, respectively), one displayed ipsilateral dominance during movements with both hands (S4), and one solely displayed contralateral dominance (S3). With their left hand, S6 displayed the only instance of bilateral dominance of the four participants. Overall, the ipsilateral dominance was split evenly between the movements with dominant and non-dominant hands, and the participants displayed varying degrees of laterality. 

The distribution of the behavioral scales under the normal, at-risk, and clinical ranges are displayed in Figure 3. S3’s parent reported a level of aggression that fell within the clinically relevant range and seven scales within the at-risk ranges. S3 reported three scales that fell within the clinical ranges (sensation seeking, school problems, and somatization) and six behavioral scales within the at-risk ranges. S4’s parent reported three scales that fell within the clinically significant ranges (anxiety, somatization, and internalizing problems) and seventeen scales that fell within an at-risk range. S4 reported three scales (anxiety, a sense of inadequacy, and attitude to school) that fell within the clinical ranges and six ranges that fell within the at-risk ranges. S5’s parent reported a level of somatization that fell within a clinically relevant range and seven scales that fell within an at-risk range. S5 also reported a level of somatization that fell within a clinically relevant range. S6’s parent did not report any behavioral scales that fell outside of the normative ranges, while S6 reported two scales that fell within the at-risk ranges. Additionally, all participants and parents scored within acceptable ranges for the validity scales.

## 4. Discussion

Overall, this study found that all the participants had lower manual dexterity scores compared to their age- and sex-matched norms, which supported the researchers’ hypothesis. Three of the participants also displayed ipsilateral-dominated activation in the motor cortex, partially supporting the hypothesized results based on the theory of neuronal group selection. Additionally, three of the participants had scores that were in the clinical or at-risk ranges, partially supporting the hypothesis of the researchers. Taken together, these results suggest that isolation can lead to negative impacts in pediatric patients, but further research should be conducted to further control for the effects of hospitalized isolation. For future studies, longitudinal data should be collected with participants completing assessments before, during, and after periods of isolation. Additionally, this information can be collected across a hospital department or system to better understand the impacts of the specific isolation protocols in place. Such interventions might aid in detecting potentially deleterious impacts which might otherwise be masked after discharge from hospital while still in at-home social isolation [28,37,38,39]. While motor dexterity has been understudied in children following isolation, previous studies have found lower levels of motor dexterity in pediatric populations without chronic illness, which was coupled with cortical structural changes [40]. The results of the current pilot study suggested that the abnormal ipsilateral hemispheric control and gross hand dexterity deficits found in children experiencing hospitalized isolation could remain for as long as 60 days after the initial isolation period. There was no evidence of interlimb transfer in the scores obtained between the first and second hand used for the task. This may have been associated with the practice period, which was integrated prior to the first task to ensure that all the participants understood the directions. Previous investigations have demonstrated that the contralateral hemispheric control of the upper limbs is the standard configuration when performing motor tasks in typically developing individuals [41]. The degree of ipsilateral dominance found in this study has not yet been reported in chronically ill children after isolation. As the laterality index is defined as a ratio of activity between the hemispheres, ipsilateral dominance does not preclude the activity in the contralateral region of interest. The participant who displayed ipsilateral dominance on both hands (S4) was undergoing hospitalized isolation at the time of the data collection for chemotherapy treatment. He was the only participant under hospitalized isolation at the time of collection and was also the only one to be undergoing chemotherapy treatment, meaning it was not possible to differentiate these two factors with the given information. There is a strong possibility that this participant‘s hemispheric laterality was impacted by the medicines he was being given through this treatment, which can suppress cortical activity [42]. The isolation period itself may have also exacerbated the impacts of this medicine, leading to the more evident ipsilateral dominance observed. To better understand the factors that may have contributed to these neuroplastic changes, anatomical and functional magnetic resonance imaging (fMRI) can be carried out in the future. The fMRI can be utilized to study the wider and deeper regions of the brain than the current setup to examine the potential widespread range of these functional differences. Imaging of the frontal lobe would also be relevant. Additionally, the anatomical MRI can be used to examine the scale at which these neuroplastic changes are occurring in order to gain a better understanding of the speed at which adaptation to isolation might occur. Electroencephalograms (EEGs) are widely used to monitor pediatric patients in hospitals, particularly in specialized intensive care units, and would be useful to leverage for monitoring the impacts of isolation [43]. Quantitative EEGs (qEEGs) may be better suited for this process as they are able to capture long-term changes more effectively compared than typical EEG techniques [43].

Three out of the four participants displayed behavioral scales that fell within the clinical ranges in both their self-reports and parent reports. Previous studies with chronically ill pediatric populations have found higher occurrences of somatization, anxiety, and depression compared to their control groups [44]. These results were further supported by the findings of this study. The participants in this study did have more scales falling into the at-risk and clinically relevant categories than would be expected in children of their ages based on normative data. The BASC-3 tests contain validity scales to control for the impact of the raters, for both self-reports and parent reports. All participants and parents were deemed to be within acceptable ranges, which indicates that the readers were consistent between questions on similar scales, and that they were not overly positive or negative in their scores. The behavioral scores do not constitute a diagnosis of any condition, as diagnoses are only made by qualified clinicians after specific processes in accordance with their ethical and professional regulations. However, the extent of these neuroplastic adaptations to isolation should be further explored to ascertain whether these changes are associated with deficits in other areas of functioning for pediatric patients. Mental healthcare for children should be proactive and integrated throughout the treatment period, and in turn might help prevent other functional deficits [45]. Amongst the areas which might impact motor skills are somatization scores, which were among the most commonly reported within the at-risk and clinically relevant ranges. Many of the questions contributing to the somatization scale asked about the degree to which children felt sick or dizzy, but it did not include fatigue. As children in hospitalized isolation frequently receive medical treatments or check-ins at various times of day, fatigue should be included in future studies to understand which specific behavioral indicators could be utilized to prevent motor skill deficits. However, future longitudinal studies can continue using the BASC-3 assessment as it has been validated through to 21 years of age. The long-term impacts of hospitalized isolation on behavior are not well understood despite them being critical to monitor and to address to better serve the needs of patients who are experiencing this isolation.

The main limitations of the current study include a limited sample size and the retrospective nature of the data collection. It also includes a wide range of ages for pediatric patients, which comprise a variety of developmental changes that may be anticipated and should therefore be carefully controlled for in future studies. These factors make it difficult to generalize the results as a direct result of isolation. Another major limitation in this study was failure to obtain a comprehensive medical history of the participants. This should be included in future research, as some medications may impact the cortical activities or gross manual dexterity. Future studies should also include at least one control group for comparison. This study examined the impacts of isolation, and it took place during the COVID-19 pandemic, during which potential members of the control group had been placed under directives for social isolation. At the time of data collection, these directives varied greatly within the local community to the extent that some potential healthy control participants would be attending school in person daily while others would be isolated at home with only their immediate family full time. Therefore, the heterogeneity of isolation was deemed to be inappropriate to serve as a control group. It is also noteworthy that during this period of mandated social isolation, many individuals who could have served in the control group were also exhibiting symptoms of decreased levels of mental health through this heterogeneous social isolation [46,47]. Comparisons to a group of control participants undergoing isolation due to COVID-19 were expected to confound the results of this study as both groups would be undergoing limited socialization, although in different contexts, and so they were excluded from the current study. 

## 5. Conclusions

All assessments used in the current study are highly portable and can be administered within the patient’s hospital room during a period of isolation. This methodology can be used to assess whether dexterity or behavior interventions are necessary on an individual basis for pediatric patients. The current study serves to establish a methodology that can be used to monitor and direct interventions for pediatric patients undergoing isolation due to chronic illness along with matched control participants. Studies should also examine the long-term impacts of isolation to determine whether reintegration into socialization reverses the impacts in either the motor or behavior aspects of isolation.

## Figures and Tables

**Figure 1 children-10-01287-f001:**
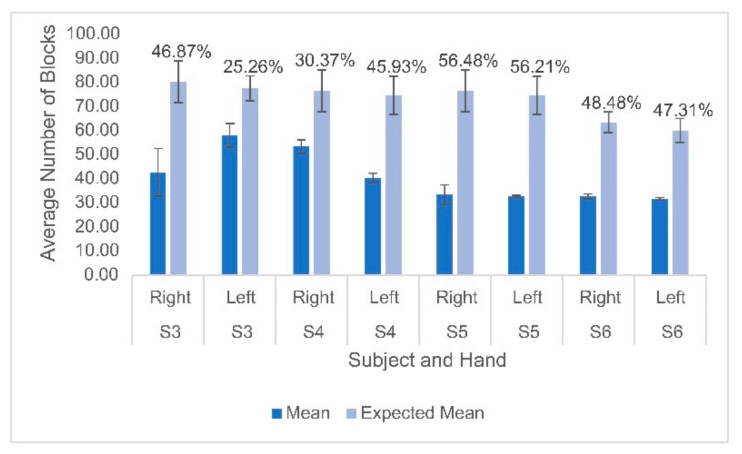
Mean and standard deviation of the number of blocks moved for each participant on both hands compared to the expected means based on normative data. Percentages displayed above the bars indicate the difference between the obtained and expected values for each participant’s hand. Lighter bars indicate the predicted values based on normative data and darker bars indicate the actual values.

**Figure 2 children-10-01287-f002:**
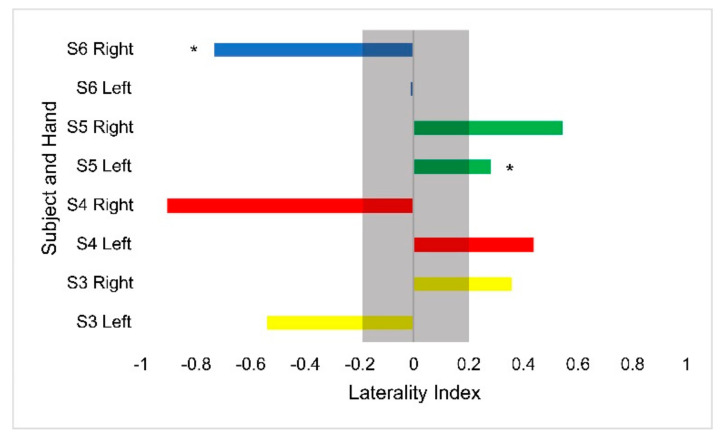
Laterality index values for the Box and Blocks task by hand for each participant. Asterisks denote ipsilateral activation. The gray bar outlines the region of bilateral dominance. Each participant’s scores are displayed in a different color.

**Figure 3 children-10-01287-f003:**
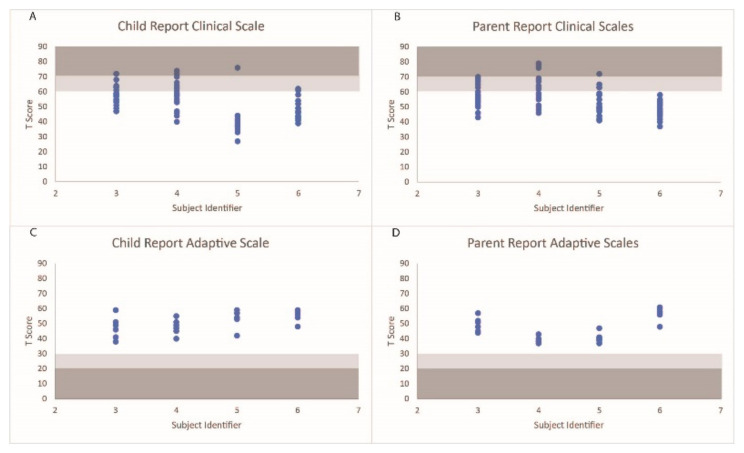
T scores of the behavioral scales on the self-report and parent reports for each participant. Scale ranges falling within the at-risk categories are shaded in light gray, and ranges falling within the clinically relevant ranges are shaded in dark gray.

**Table 1 children-10-01287-t001:** Subject demographic and clinical characteristics.

Subject	Age (Years)	Sex	Dominant Hand	Length of Most Recent Stay	Reason for Hospital Stay
S3	16	Male	Right	7–8 Days	Infection due to leukemia
S4	14	Male	Right	2 Days	Chemotherapy for osteosarcoma
S5	14	Male	Right	17 Days	Rejection
S6	9	Male	Right	27 Days	Bone marrow transplant

## Data Availability

The data presented in this study are available on request from the corresponding author. The data are not publicly available due to privacy concerns.

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
