# Peer review of "Low Motor Dexterity and Significant Behaviors Following Hospitalized Isolation in Children"

_children, 2023, doi:10.3390/children10081287_

Round 1

Reviewer 1 Report

The paper by Kaitlin Fraser et al. is a very interesting study evaluating the impacts on motor performance and socio-behavioural profile in young people undergoing isolation for serious health problems. It is a paper that offers very important insights (also considering the recent Covid-19 pandemic) into the consequences of isolation and the possible mechanisms responsible, with considerable methodological rigour. The sample is very small and heterogeneous, which precludes the generalisability of the results, but it is still possible to formulate hypotheses on the basis of the data obtained. I only have a few suggestions for the Authors: 

- In the introduction, the Authors rightly point out that social isolation is different from hospital isolation and then specify that all recruited patients 'had undergone hospitalised isolation or were still in social isolation at the time of the data collection'. Were there differences between these two situations in the results?

- The duration of hospital isolation varied widely among the patients involved (i.e. from 2 to 27 days). Of note, S6 (the patient with the longest isolation) is also the only one characterised, using the left hand, by instances of bilateral dominance among the four participants. What considerations could the Authors draw from this finding?

- The LI is certainly a good parameter for the Authors' purpose. However, it would also be interesting to assess whether there is a difference between different regions of the most involved hemisphere. For example, if it is the complexity of the task that motivates the differences in activation between the two hemispheres, are the frontal regions (typically related to high cognitive load) more active than the posterior ones in those with different hemispheric activation patterns than in healthy subjects?

- It would be interesting to assess, also in view of these different hemispheric activations, the presence of an Interlimb transfer. Did the Authors find differences in performance in the second group of Box and Block Test tasks (i.e. with the second hand used according to randomisation) in those who started with the dominant hand compared to those who started with the non-dominant hand?

- All recruited patients are characterised by extremely debilitating clinical situations and therapies. Did the Authors use any standardised scale to assess fatigue and its impact on performance? If not, it would be worth considering its impact on performance 

- Chemotherapies can have serious effects on patients' cognitive performance. In fact, S4 is the only one with ipsilateral dominance during movements with both hands. What might be the implications?

- The ages of the patients are very different and this could have affected the results obtained, both from a motor and behavioural point of view. I suggest the Authors make some considerations in this regard. 

- I suggest the Authors a small table summarising the clinical and demographic characteristics of the patients recruited.

- Is there EEG data on the consequences of isolation? Given the differences between NIRS and EEG, I suggest adding a consideration in the Discussion in this regard. 

- Line 46: "sequestered" is probably too strong a term, I suggest changing it to, e.g., "confined"

- Line 67: probably missing a full stop and/or citation.

Reviewer 2 Report

First of all, I would like to congratulate the authors for the study carried out. It is a very interesting and well-conducted study. 

The theoretical contextualisation is developed correctly and in a very orderly manner. I would suggest updating the citations a bit more using only the year 2020 onwards. 

I would also recommend dividing the Material and Method section into the following sections: Design, Participants, Instruments and Variables, Data Analysis and Procedure. 

Otherwise, the article shows completely reliable and significant data in the field of study in which it is framed. 

Round 2

Reviewer 1 Report

The Authors have properly addressed all comments and the paper, in my opinion, is now suitable for publication

Reviewer 2 Report

The article has been improved following the proposed recommendations. I agree to the publication of the article.